# An Out-of-Plane Operated Soft Engine Driving Stretchable Zone Plate for Adjusting Focal Point of an Ultrasonic Beam

**DOI:** 10.3390/s19183819

**Published:** 2019-09-04

**Authors:** Guo-Hua Feng, Hong-Yu Liu

**Affiliations:** Department of Mechanical Engineering, National Chung Cheng University, Chiayi 621, Taiwan

**Keywords:** zone plate, ultrasonic beam, ionic polymer, actuator

## Abstract

This paper presents a soft engine which performs up-and-down motion with four planar film-structured ionic polymer—metal composites (IPMC) actuators. This soft engine assembled with a stretchable Fresnel zone plate is capable of tuning the focus of ultrasonic beam. Instead of conventional clamps, we employ 3D printed frame pairs with magnets and a conductive gold cloth to provide an alternative solution for securing the IPMC actuators during assembly. The design and analysis of the zone plate are carefully performed. The zone plate allows the plane ultrasonic wave to be effectively focused. The motion of IPMC actuators stretch the metal-foil-made zone plate to tune the focal range of the ultrasonic beam. The zone plate, 3D frames and IPMC actuators were fabricated, assembled and tested. The stiffness normal to the stretchable zone plate with varied designs was investigated and the seven-zone design was selected for our experimental study. The force responsible for clamping the IPMC actuators, controlled by the magnetic attraction between the fabricated frames, was also examined. The driving voltage, current and resulting displacement of IPMC actuation were characterized. The developed soft engine stretching the zone plate to tune the focal point of the ultrasonic beam up to 10% was successfully demonstrated.

## 1. Introduction

Over the last decade, electroactive polymers (EAPs) have garnered significant attention because of the promising characteristics they exhibit in their application to soft actuators [1,2,3,4]. Among various EAPs, piezoelectric polymers such as Polyvinylidene fluoride (PVDF) have been broadly studied owing to its reliable electro-mechanical conversion property and reasonable sensing accuracy. Nevertheless, its actuating performance is very limited without a dominantly high voltage. ionic polymer–metal composite (IPMC) materials possess exceptional advantages for usage in soft actuators because they can be manipulated with relatively low voltages (usually several volts) and can produce large bending motions [5,6,7,8]. 

A common IPMC actuator is made up of an ion-exchange thin membrane with metal particles plated on its top and bottom surfaces as electrodes which form a sandwich structure [9,10,11]. The widely investigated ion-exchange membrane was Nafion. [12,13,14]. The nanochannels formed in ionic regions allow hydrated cations to migrate. When a voltage drop is applied to the electrodes of an IPMC actuator, the cations along the water molecules travel towards the cathode and gather close to the electrode interface. The anions are secured within the clusters formed from the polymer network. This causes a change in the internal stress gradient of an IPMC actuator in the direction perpendicular to its electrode surfaces, and results in a bending moment to produce a curvature on the actuator [15,16,17,18].

Researchers often use clamps to mechanically secure the connections and transmit electricity to IPMC actuators due to the difficulty that arises in connecting wires with soldering. These bulky clamps make a compact IPMC working system challenging. These difficulties can be resorted by utilizing 3D printed frames with magnetic attraction and a gold-plated conductive cloth to provide clamping pressure and electrical pads. Additionally, multiple IPMC actuators can be connected in parallel to generate greater force to produce an out-of-plane long-range motion. The implemented IPMC actuator based soft engine will be demonstrated to control the focus of the ultrasonic beam. 

High-intensity focused ultrasound (HIFU) is a non-invasive therapeutic technique that utilizes 0.5–10 MHz ultrasonic waves to cause controlled, localized tissue damage [19,20,21,22]. The focus of a typical HIFU transducer is fixed at the geometric center. The ability of adjusting focus is a definite requirement for practical usage. Tuning of the focus in the existing HIFU technique could be initiated by phased array transducers or by manipulation of the mechanical translation stage [23,24,25]. However, the complicated electronics and high manufacturing costs would be clear disadvantages. Hence, we propose to study a flexible and stretchable Fresnel zone plate to adjust the focus of an ultrasonic beam using the developed IPMC soft engine (Figure 1).

## 2. Theoretical Study of Zone Plate

### 2.1. Ultrasonic Wave Diffraction

Based on the analysis of the diffraction of the ultrasonic waves which are excited by a cylindrical plane piston transducer with an area S, a Green’s function could be derived [26,27,28]. Along with the Rayleigh-Sommerfeld formula, we have a pressure expression p at any point (z ≥ 0) in a liquid medium (Figure 2a):$$p=\omega^{2}\rho_{m0}\phi$$
(1)$$\phi \left(x,y,z\right)=\int_{S}^{}\phi \left(x\prime ,y\prime ,0\right)\frac{2z}{R}\left(jk+\frac{1}{R}\right)\frac{e^{-jkR}}{4\pi R}dS$$
where
$$R=\sqrt{{\left(x-x\prime \right)}^{2}+{\left(y-y\prime \right)}^{2}+z^{2}}.$$

With the following assumption,
$$\sqrt{{\left(x-x\prime \right)}^{2}+{\left(y-y\prime \right)}^{2}}<<z$$

The Green function has another solution which yields the potential function ϕ at any point (z≥0) in a liquid medium. This is
(2)$$\phi \left(x,y,z\right)=-\frac{1}{2\pi }\int_{S}^{}u_{z}\left(x\prime ,y\prime ,0\right)\frac{e^{-jkR}}{R}dS$$
where $u_{z}$ is the displacement in the z direction. We assume that the displacement at the surface of the transducer $u_{z}\left(r,0\right)=u_{0}$ is uniform over its radius r=a and zero outside it. Thus, the amplitude of a wave at point z on the central axis of the transducer is
(3)$$\phi \left(0,z\right)=-u_{0}\int_{r\prime =0}^{a}\frac{e^{-jkR}}{R}r\prime dr\prime ={\frac{u_{0}e^{-jkR}}{jk}|}_{r\prime =0}^{r\prime =a}$$
using the relations $R^{2}={r\prime }^{2}+z^{2}$ and Rdr=r′dr′, we have
(4)$$\phi \left(0,z\right)=\frac{u_{0}}{jk}\left[e^{-jk{\left(a^{2}+z^{2}\right)}^{\frac{1}{2}}}-e^{-jkz}\right]$$

In addition, the displacement at point z on the central axis of the transducer can be expressed as
(5)$$u_{z}\left(0,z\right)=\nabla \phi \left(0,z\right)=u_{0}\left[e^{-jkz}-\frac{z}{{\left(a^{2}+z^{2}\right)}^{\frac{1}{2}}}e^{-jk{\left(a^{2}+z^{2}\right)}^{\frac{1}{2}}}\right]$$

### 2.2. Zone Plate Design

Utilizing a zone plate with the diffraction effect of ultrasonic waves allows us to direct the planar ultrasonic wave towards a focused ultrasonic wave. A zone plate is composed of several radially symmetric rings as zones. Zones manipulate the blocks and the transmission of ultrasonic waves and hence, the resulting ultrasonic wave constructively interferes at the desired focus.

The focal length is defined as the distance between the center of the zone plate and the designed focal point. The formula for zone radii for a zone plate with focal length f is derived in the following manner:

Consider an ultrasonic source point in the plane of the zone plate located a distance r from the center of the zone plate (Figure 2b). The distance between the source point and the focus can be described as the wave path length l. When the difference between the wave path length of a source point and the length f is within one half of the wavelength, the waves interfere constructively. The wave path length satisfies the relation below, resulting in constructive interference.
(6)$$\frac{\left(n-1\right)}{2}\lambda <l-f<\frac{n}{2}\lambda$$
n=1,2,3…N
where *n* represents the zone number. While n equals either odd or even numbers, the sources in these wave path lengths arises due to constructive interference. However, the sources in wave path lengths between odd and even numbers of n cause destructive interference. 

Thus, we can define the borderline $r_{n}$ of every zone for a zone plate based on the following equations:(7)$$l-f=\sqrt{r_{n}^{2}+f^{2}}-f=\frac{n}{2}\lambda$$
(8)$$r_{n}^{2}=n\lambda \left(f+\frac{n\lambda }{4}\right)$$

If nλ/4<<f, we can use $r_{n}≅\sqrt{n\lambda f}$ for our zone plate design. 

In this study, the design restriction for our zone plate was that the outer edge radius of the plate should be limited to 9.5 mm. Under this requirement, the zone number was investigated from amongst 5, 7, 9 and 11. Also, the ultrasonic source utilized a planar-type transducer with an operating frequency of 5 MHz. Based on the acoustic velocity of water (1480 m/s), the wavelength under investigation is 296 µm. Table 1 lists the focal lengths and zone radii of four different zone number designs. The designed focal ranges according to the equation above are 61.3 mm, 43.8 mm, 34.1 mm and 27.9 mm for *n* = 5, 7, 9 and 11. A high zone number results in a smaller focal length. In the four studied cases, the value *nλ*⁄4 = 0.814 mm for the worse case (*n* = 11) was lesser than *f* = 27.9 mm. Thus the approximation, $r_{n}≅\sqrt{n\lambda f}$ was suitable. For our zone plate design, we choose the even number zone regions to allow planar ultrasonic waves to pass through. The blockage of the center zone forces the IPMC actuators to contact this region and stretch the fabricated zone plate so that the radius of each zone would be tuned.

We will experimentally verify that our fabricated stretchable zone plate could allow the planar ultrasonic wave to focus at the designed focal spot under the condition of the zone plate without any stretch. Further, while the zone plate is deformed by the IPMC actuator, the same planar ultrasonic waves will alter the focal points as a function of the zone plate stretched at different levels.

### 2.3. Analysis of Focused Ultrasonic Waves for the Designed Zone Plates

A constraint on the outer edge radius of the zone plate for our design was created. The focusing effect on varying the zone numbers within a specified dimension was investigated. The intensity distribution of ultrasonic waves along the axis normal to the center of first zone was evaluated. According to Equation (7), the resulting displacement along this axis can be formulated as:(9)$$\begin{array}{l} u_{z,total}\left(0,z\right)=u_{0}\{\left[\frac{z}{{\left(a_{2}^{2}+z^{2}\right)}^{\frac{1}{2}}}e^{-jk{\left(a_{2}^{2}+z^{2}\right)}^{\frac{1}{2}}}-\frac{z}{{\left(a_{1}^{2}+z^{2}\right)}^{\frac{1}{2}}}e^{-jk{\left(a_{1}^{2}+z^{2}\right)}^{\frac{1}{2}}}\right]+\ldots \\ +\left[\frac{z}{{\left(a_{n}^{2}+z^{2}\right)}^{\frac{1}{2}}}e^{-jk{\left(a_{n}^{2}+z^{2}\right)}^{\frac{1}{2}}}-\frac{z}{{\left(a_{n-1}^{2}+z^{2}\right)}^{\frac{1}{2}}}e^{-jk{\left(a_{n-1}^{2}+z^{2}\right)}^{\frac{1}{2}}}\right]\} \end{array}$$
where *n* is equal to 4, 6, 8 and 10 in this study.

Since the intensity of the ultrasonic wave is proportional to the square of its displacement amplitude, the resulting ratio of the intensity along the axis normal to the center of the first zone to the intensity of planar wave can be obtained by
(10)$$\frac{I\left(0,z\right)}{I_{0}}={\left|\frac{u_{z,total}\left(0,z\right)}{u_{0}}\right|}^{2}$$

Figure 3 shows the calculated intensity ratios of four kinds of zone plate designs. The results show that the focal point with the maximum intensity is as similar to our design (Table 1). The magnitudes of maximum intensities exhibit an increasing trend from approximately 35 to 135 while the zone number rises from 5 to 11. 

### 2.4. Actuator Consideration 

To drive the flexible and stretchable zone plate, a small size and uncomplicated structured actuators is necessary. In addition, working in a water environment would be a benefit for ultrasonic wave transmission. An actuator made of electroactive polymer material would be a good choice. The electroactive polymers based on the light intensity or temperature change as actuation mechanism could not be better than the electrical control for actuation in this study. This is because electrical input could be easily provided by wire transmission, while guiding light emission or adding heat source to the actuator could cause extra assembly difficulties. 

Compared to some electroactive polymers operated in high voltage, the ionic EAP actuators usually operated at several volts could be an advantage. The promising ionic EAPs include IPMC, conducting polymers and nanostructured carbon-based materials. The commercial available conducting polymer such as Poly (3,4-ethylenedioxythiophene): poly (styrenesulfonic acid) (PEDOT: PSS) has high flexibility and excellent thermal stability [29]. The nanostructured carbon-based device such as the graphitic carbon nitride nanosheet electrode-based ionic actuator displays high electrochemical activity and large specific capacitance [30]. However, IPMC would have better electromechanical conversion efficiency compared with conducting polymer actuators and better force output compared with nanostructured carbon-based ionic actuators [4,31,32,33]. This justifies the IPMC actuator as a good candidate for this study.

Although the position holding capability of soft ionic actuators could be an issue, the goal of this study was to change the focal distance of ultrasonic beam as a function of time. We aimed at generating an ultrasonic beam of time-varying focus applied to the object with periodically volume change. For example, if a specific location of an animal heart was required to be treated with HIFU, this time-varying focused ultrasonic beam could be an efficient scheme because the heart beats periodically.

## 3. Device Design and Fabrication

Stereolithography was used initially to fabricate a pair of octagonal frames for integrating the magnets, the zone plate and IPMC actuators. The 3D printing machine was manufactured by XYZ printing Co. with model no. Nobel 1.0 A (Taiwan). It has an XY axis resolution of 130 µm and layer thickness setting of 25/50/100 µm. The photopolymer resin was provided by the same manufacturer. Solidworks software was used to plot our design structure and translate into STL format to perform the printing task. The layer thickness was set to 25 µm to obtain the best resolution. 

Regarding the frame structure, the inner diagonals of the octagonal frames were taken as 24 mm and each inner side was 9 mm (Figure 4). The width of the frame was 5 mm and each outer side length was 13.25 mm. For the top surface of the upper frame and the bottom surface of the lower frame, each had four symmetrically arranged circular cavities with a diameter of 4 mm to place the circular magnets. The bottom surface of the upper frame had 4 wedge-shaped protrudes and four small cavities to fit the four slots and four small pillars on the top surface of the lower frame, respectively. The four protrude-slot pairs were designed to anchor the IPMC actuators and transmit the necessary driving electricity. The wedge-shaped protrusion allows IPMC actuators anchored to the 3D structure was not simply in a horizontal plane. This allows the endpoints of the actuators with a tilting upward arrangement in the initial state to just touch the center ring of the zone plate, which effectively utilizes IPMC actuators. The four small cavity–pillar pairs were designed to ensure that the upper and lower magnets were properly aligned to attract each other so as to maintain the bonding force between the upper and lower frames during device operation. Also, the positioning holes made on the stretchable zone plate were fixed to these four pillars in order to affix the zone plate at the center region of the frame structure during assembly. 

The IPMC actuator fabrication followed [34,35]. Starting with a large 190 µm-thick Nafion-117 membrane (DuPont Co., USA), it was cut to a suitable size for easy processing. The sliced membrane was immersed into diluted hydrochloric acid (20%) for 5 h at 50 °C. This expanded the pores on the surface of the Nafion membrane to facilitate the platinum particle precipitations. After rinsing the processed membrane with DI water, the platinum salt solution was prepared with 0.06 g tetraamine platinum (II) chloride hydrate [Pt(NH_3_)_4_]Cl_2_∙xH_2_O] and 30 mL DI water. The Nafion membrane was soaked in this chemical solution for about 12 h at 50 °C. While the edges of the Nafion membrane displayed a light-yellow color, the chemical reduction was performed by 0.05 M LiBH4 solution at 50 °C. The platinum particles covered both sides of the Nafion membrane to form the electrodes. After ion-exchange with lithium hydroxide for 8 h, the IPMC membrane formation was complete. Each actuator was sliced 15 mm long and 2 mm wide. The designed working length was 10 mm, i.e., 5 mm at the end of the strip clamped by the protruded/slot pair of the frames.

The unique zone plate made up of metal foil with a flexible structure not only allows us to focus the planar ultrasonic wave at the designed focal point but also enables us to change each zone radius of the zone plate to tune the focal point. The designed rings of the zone plate and the associated ultrasonic wave focusing characteristics have been discussed above. To link the neighboring rings of the zone plate to each other and ensure flexibility, we included four symmetrical serpentine-shaped structures to connect two neighboring rings. The serpentine-shaped structure has been studied and implemented for stretchable electronics applications [36,37,38]. For instance, the sinusoidal stripe was specified for the serpentine-shaped structure with its period, amplitude and width denoted as L, A and W, respectively. If W/L = 0.1 and A/L > 0, the stripe can benefit from both in-plane bending and out-of-plane twisting. Hence, the maximum strain of the stripe would be less than 0.02 mm for relative elongation at both ends reaching 25% [39]. When the stripe was repeatedly bent, twisted, and stretched, it had no appreciable fatigue.

Instead of the sinusoidal shape design, we employed different serpentine-shaped structures for our previous work and demonstrated considerable elongations with a lower effective spring constant [40]. In this study, the flexible structures connecting the neighboring rings of the zone plate were chosen. Figure 5 shows the photomask. For each zone plate design of four varied ring numbers, we exploited the radius difference between the first and second zone as one period of serpentine curve, i.e., $p=r_{2}-r_{1}$ and divided the length p into 4 equal segments with the coordinate of each node (x0, y0), (x0 + 0.25p, y0), (x0 + 0.5p, y0), (x0 + 0.75p, y0), (x0 + p, y0). Next, two points (x0 + 0.25p, y0 + 0.5p) and (x0 + 0.75p, y0 − 0.5p) as the maximum amplitude of the serpentine curve were marked and AutoCAD drawing functions to connect (x0, y0), (x0 + 0.25p, y0 + 0.5p), (x0 + 0.5p, y0), (x0 + 0.75p, y0 − 0.5p), (x0 + p, y0) to a spline line were applied. A 0.5 mm offset was employed on both sides of the spline line as the boundary line of the designed serpentine structure. Once the serpentine structure was ready, we copied it to connect it to other neighboring rings. 

Since the zone regions became gradually smaller outwards, the designed serpentine structure became larger than the connected transparent space. Hence, a rectangular retraction region was created to give enough space to fit the serpentine structure. The retraction region would gradually extend outwards causing the remaining ring portion to shrink. A critical dimension should be specified to allow the remaining ring portion at retraction region preserving a necessary stiffness so as the shrinking ring portion without significant deformation or distortion during the serpentine structure stretch. We set the critical width of the shrinking ring portion to 0.3 mm by experimental trial. If the serpentine structure was larger than the fitting space with the critical width shrinking to 0.3 mm for a designed portion of the zone plate, the center point of the designed serpentine structure would then be aligned to the middle point of the line segment between the ring edge and the rectangular retraction edge. The serpentine structure that overlapped with the ring portion was removed to just fit the space.

To fabricate the zone plate, we started with a 10 µm-thick square titanium foil as the substrate followed by standard photolithography with the abovementioned photomask and wet etching. The resulting zone plate showed very little residual stress effect on the fabrication process and the whole zone plate exhibited a flat surface. 

Finally, the assembly process was executed. The gold-plated conductive cloth was used as the electrode pad material. It was then cut into small strips to fit the widths of wedge-shaped protrudes and slots of the frames. These conductive strips were then individually bound around the wedge-shaped protrudes and slots of the frames as the electrode pads. The NdFeB magnets were individually embedded into the circular cavities of both frames with proper polarization direction. The four positioning holes of the zone plate were aligned and mounted to the four pillars on the top surface of the lower frame. The actuation module of four IPMC beams was positioned to allow each IPMC beam to be arranged at slots of the lower frame. After joining the processed upper and lower frames together, the soft engine made of IPMC actuators and a zone plate was complete for testing (Figure 6).

## 4. Device Characterization and Discussion

### 4.1. Flexibility and Stiffness of Fabricated Zone Plate

To examine the property of the fabricated zone plate being able to stretch in the out-of-plane direction, the flowing experiment was conducted. We utilized the 3D FDM printing technology to construct a L-shaped holder with a hexagonal opening of diagonal dimension 9 mm in the long side (Figure 7a). The fabricated zone plates with 4 different designs were investigated. The edges of the zone plates were individually secured on the bottom side of the printed octagonal opening with 3 M double-sided tape during testing. Moreover, a small pillar stand with a diameter of 5 mm and height of 20 mm was made by stereolithography. The pillar served as a force coupler with one end of the pillar stand contacting the force balance and the other end touching the lower surface of the zone plate. 

During the testing, the pillar stand was placed onto the weighing plate of a precision electronic balance. One side of the L-shaped holder was fixed on a computer-controlled z-axis precision stage to improve adjustment upwards and downwards. The L-shaped holder was moved to an appropriate position. The central disc of the zone plate just contacted the top surface of the pillar. This position is set to the zero displacement and zero force readout on the electronic balance. The stage is then set to move in the z-direction downwards in steps of 50 µm. The central disc of the zone plate is subjected to a lateral force through the serpentine structures connected between the rings of the zone plate. The central disc would also exert a downward force on the fabricated pillar causing an increase on the force readout of the electronic balance. We recorded the force readout of the electronic balance while the stage moved each step.

The stretched displacement in the out-of-plane direction of the zone plate as a function of applied forces was obtained as shown in Figure 7b. Four cases with a zone number of 5, 7, 9 and 11 were individually studied. The zone plates for all cases can be returned to its original positions after a 2 mm stretch. The experimental results show an approximately linear stiffness in the direction vertical to the zone plate and the zone plate with the lowest zone number possesses the greatest stiffness. In more detail, the slightly nonlinear phenomenon can be divided into two linear regions for a displacement range of 0–0.3 mm and beyond 0.3 mm. The five-zone number case displays an obvious large stiffness compared to the other three cases. The stiffness does not exhibit a significant variation for the seven, nine and eleven zone number cases. A lower stiffness exists for small displacement regions and reaches saturation values for large displacement regions for our test range. The stiffness of the five zone numbers case reaches about 25 gf/mm and for the rest, approximately 21 gf/mm. Based on the principle of selecting lower stiffness and larger focal length, we choose the fabricated zone plate with seven zone number for the following study.

### 4.2. Magnetic Attraction Between Frame Pair

The magnetic attraction force of the frame pairs was investigated by the experiment below: 

We secured the magnet embedded lower frame to a relatively heavy load so that the magnetic attractive force induced by the upper frame was less than the gravity of the load plus the lower frame during the explored distance between the upper and lower frames. This allowed us to understand and even control the attractive force between the lower and upper frames, which is a critical parameter affecting the gold cloth pads exerting pressure onto the electrodes of IPMC actuators during operation.

We implemented this setup by stacking regular glass slides (1 mm thick) as a heavy load. 12 glass slides were joined with double-sided tape and the lower frame was also secured to the center of the top glass slide with double-sided tape. The entire kit of the lower frame and glass slides was then put on a precision electronic balance with a zero readout. Since the maximum capacity of the electronic balance was 400 g, the weight of this entire kit should be confined within 400 g. The upper frame was mounted underneath the octagonal opening of the L-shaped holder of the z-stage and aligned above to the lower frame (Figure 8a). The distance between the two frames could be adjusted by the stage to measure the variation in magnetic attractive force. 

We began with regulating the upper frame from a distance far away from the lower frame, at which the readout of the force balance was the same as the status without the upper frame above the lower frame. After that, the z-stage was slowly moved down to the starting position to change the readout of force balance and we recorded the distance and readout. For the investigated two pairs of magnet arrangement in the frames, the measured distances between the frames were 40 mm and 55 mm with corresponding readout of 63.067 gf and 54.252 gf for the magnet diameter of 4 mm and 3 mm pairs, respectively. 

When we gradually lowered the z-stage, the readout of the electronic balance reduced because of the magnetic attractive force. For example, the readouts displayed 54.53 gf and 43.28 gf for a 1 mm distance between two frames with magnet diameters of 4 mm and 3 mm pairs respectively. The readouts decreased to 50.06 gf and 36.527 gf while the gap was only 0.1 mm. The smallest distance between the frames was characterized at 0.1 mm due to practical operation of the fabricated IPMC actuators. The thickness of the IPMC actuator was approximately 0.2 mm and two pieces of gold-cloth pads were about 0.2 mm. Hence, the gap between frames was supposed to be greater than 0.1 mm while IPMC actuators were clamped by frames in operation.

Figure 8b shows the magnetic attractive force between the upper and lower frames. The curves derived from the values of the readout without upper frame attraction are 63.067 gf and 54.252 gf for the two cases deduct the values of electronic balance readout at measured positions. Results show that the magnetic attractive force is proportional to the reciprocal of the distance between two frames. The measured forces were in the range of 16–18 gf and 12–13 gf for the gap in the range of 0.1–0.3 mm for magnet diameters of 4 mm and 3 mm. Since the four magnet pairs were symmetrically arranged around the upper and lower frames, this implied that each pair provided 4–4.5 gf and 3–3.25 gf for clamping the IPMC actuators during operation. The former pair were employed in subsequent study. 

### 4.3. Actuation Capability of Developed Soft Engine

The IPMC actuators are the critical component in the designed soft engine and their driving ability affected the out-of-plane motion of the stretchable zone plate. To evaluate IPMC performance, two experiments were conducted in water: (1) one simple IPMC actuator was driven with the designed frame setup to provide clamping pressure and an electrical signal. (2) The 4 IPMC actuators were electrically connected in parallel to stretch the flexible zone plate. The electrical property of IPMC actuators in operation were characterized in both the experiments. Details are described below.

Figure 9a shows the experimental setup for testing a single IPMC actuator. The dimension of the IPMC actuator was 15 mm × 2 mm × 0.21 mm. The portion of 5-mm-long IPMC actuator at one end was clamped by the protruded/slot pair of the frames and the rest portion of the IPMC actuator could be activated to perform the bending motion. The IPMC actuator along the frames was placed inside a glass container and water was added to allow the IPMC actuator to be fully immersed during operation. Let us assume that the water level was 10 mm above the highest point of the actuator at the initial state. We use the circuit with an external resistor of Rext = 91 Ω for measuring the current flowing and the voltage across the IPMC actuator. The square waves of 16 V peak-to-peak (Vpp) with a frequency of 0.2 Hz produced by a function generator was delivered to the circuit to drive the IPMC actuator. The data acquisition card was applied to obtain the voltage V1 and V2 with a sampling rate of 1 kHz. The actual voltage drop between the two electrodes of the IPMC actuator could be derived from V2-V1. The current was derived by V2/Rext. The laser displacement meter with a resolution of 0.1 µm was employed to measure the displacement of the cantilever IPMC actuator at its front end. Figure 9b shows the voltage, current and displacement results simultaneously as a function of time. The maximum voltage supplied to the electrodes of IPMC actuator was 6.5 Vpp (±3.25 V) based on the functional generator setting of 16 Vpp. Although this driving voltage could result in the electrolysis of water occurring, this voltage setting allowed us easily to monitor the ultrasonic beam focusing change by producing large displacement of the IPMC actuator in the subsequent study. The curve entails charging and discharging phenomenon due to the capacitance effect of IPMC. The current displayed a periodic variation with the peak current swiftly dropping from 66 mA to 33 mA (absolute value), thereby reaching a near steady-state value. This current property could be decomposed into faradic and non-faradic currents in each cycle [41,42]. The non-faradic current could be associated with current greater than the steady-state value (absolute value). The lithium cations carried along with the water molecules moving from one electrode to the opposite electrode alternatively caused the non-faradic current and the lateral strain gradient in the direction perpendicular to the electrodes. The faradic current could be considered as the constant current 33 mA (absolute value) resulting from the effect of charge transfer at the interface and mass diffusion [43]. We observed that small amount tiny bubbles emerge around the electrode region. This could be attributed to water electrolysis causing the hydrogen and oxygen gas evolution. The largest displacement was approximately 3.2 mm while reaching a steady state. The relatively larger displacement in the initial stage could be explained by the cations sparsely located inside the IPMC actuator after the ion-exchange process of fabrication. After several driving cycles, the cations were regulated to move to the bi-stable positions inside Nafion, which was symmetrical to the neutral axis of the cantilever IPMC actuator with time.

The hysteresis between the displacement and consuming current of IPMC actuator was observed. While the applied voltage polarity was switched such as from the negative to positive value, the current across the IPMC actuator immediately reached a maximum value and then dropped to a near constant value. The displacement gradually increased and attained a maximum value at the moment of the current approximately approaching the constant value. The time between the peak values of the current and displacement was about 1.5 s. 

To ensure the repeatability of the IPMC actuators during operation, the fabrication process control was important. It was worthwhile to mention that a sufficient time to immerse the Nafion membrane into the platinum salt solution would help the subsequent platinum participation on both surfaces of Nafion membrane. A 12 h treatment could produce very similar actuation results according to our experience. Based on this fabrication process treatment, we measured the displacement of the fabricated IPMC actuator to investigate its working stability. The actuator was tested by continuously driving for 3000 s (600 cycles), which was much longer than the necessary time for the subsequent experiments. We then examined the displacement of the IPMC actuator operated in the first cycle and last cycle and found that the displacement decrease was within 15% after this long-period operation.

The soft engine of the frame pairs clamping four IPMC actuators was integrated with the stretchable zone plate to place the water tank for testing (Figure 10a). The water level was also approximately 10 mm above the top surface of the zone plate. Four electrical pads on the lower frame were wired together, just like the upper frame. Then the wires bonded to the upper and lower pads were connected to the same driving circuit. The laser displacement meter was used to measure the displacement at the center of the zone plate. The 16 Vpp square wave at 0.2 Hz with zero bias from a function generator was simultaneously given to drive the 4 IPMC actuators. 

The consumed total current and the voltage drop of the four-actuator module was acquired by the same circuit described above instead of the single actuator replaced by the four actuators connected in parallel (Figure 10b). The actual driving voltage across both the electrodes of the IPMC actuator was equal to that of the single actuator (6.5 Vpp). The peak current flowing through the four actuator module was 82 mA and the steady state value was 40 mA (absolute value). The peak-to-peak magnitude of measured displacement displayed approximately 1 mm while reaching a steady-state. The displacement was about one-third of the displacement for a single IPMC actuator without any external loading for the same voltage actuation. The lowered displacement could be expected due to the stiffness in the stretchable zone plate in its out-of-plane direction. Moreover, the total current flowing through the four actuators was larger than a single actuator, which was reasonable because of the parallel connection, which reduced the effective resistance. Interestingly, if we consider the average current distributed in each actuator, the current in each could be 20.5 mA and 10 mA for peak and steady-state value. The two critical values were approximately one-third compared to the current of a single actuator. This ratio also reflected the IPMC actuator’s displacement with zone plate loading compared to that without zone plate loading.

### 4.4. Driving Stretchable Zone Plate to Tune the Focal Point of Ultrasonic Beam

To examine the performance of the developed soft engine operating the stretchable zone plate to tune the focus of an ultrasonic beam, a 5 MHz commercially available planar ultrasonic transducer, hydrophone, and tested device were setup inside a water tank (Figure 11a). The distance between the flat surface of the ultrasonic transducer and the tested device was set as 1 cm. The four IPMC actuator connected-in-parallel module was driven with the same circuit as described in the previous section. The function generator was set at a 0.2 Hz square wave of 16Vpp and 20Vpp to actuate the four IPMC modules, respectively. This allowed the zone plate to be stretched towards the ultrasonic transducer and bounce back repeatedly. The maximum displacements were measured as 1 mm and 2 mm, respectively, for 16 Vpp and 20 Vpp settings. The hydrophone was aligned and anchored in a holder so that the sound pressure measurement point could be moved along the center line of the ultrasonic transducer. 

The zero point in the measurement was set as the initial plane of the zone plate. We then adjusted the gauging position away from the ultrasonic transducer in steps of 2.4 mm. The maximum amplitude of the hydrophone output voltage was recorded from the oscilloscope for each step. Results showed that the zone plate effectively focused the ultrasonic beam and that the focus could be adjusted through the stretch of the zone plate (Figure 11b). The focal point could be tuned from 45.6 mm at the zone plate without any stretch to 40.8 mm within the operated soft engine. The tunable range of 4.8 mm is thus about 10% of its original focal distance.

The developed IPMC actuator based engine could further enhance its function in the future. For example, the zone plate was held at a specific position by controlling the IPMC actuators so the focused ultrasonic beam could be applied at a stationary location for a certain time. Because of the nonlinear manipulation mechanism of IPMC actuator, the position holding could not simply provide a constant voltage. Some control schemes such as adjusting the time interval of controlling signal or varying the amplitude of control signal would be employed.

## 5. Conclusions

An innovative soft engine actuated by flexible IPMC actuators to execute the long-range out-of-plane motion is proposed, fabricated and tested. The fabricated IPMC actuator module along with the micromachined stretchable zone plate and stereolithography constructed upper and lower frame pairs successfully comprise the soft engine and make it functional. Besides, the fabricated zone plate effectively makes the planar ultrasonic beam focus at the designed point, and the engine’s dynamic adjustment of the focal point of the ultrasonic beam by driving the stretchable zone plate is demonstrated. 

## Figures and Tables

**Figure 1 sensors-19-03819-f001:**
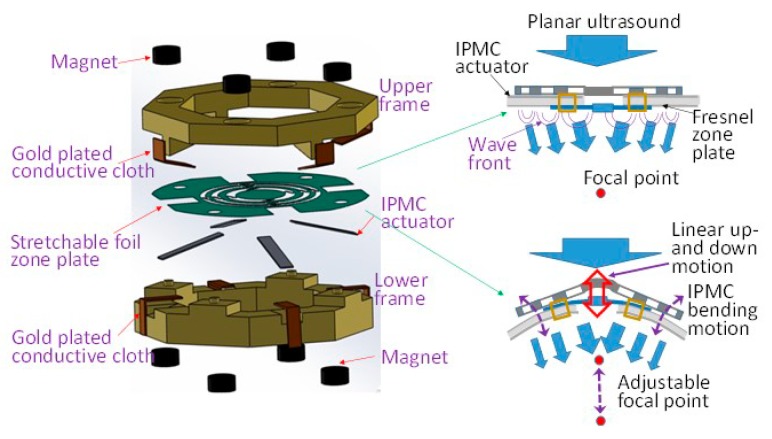
Concept of ionic polymer–metal composite (IPMC)-based soft engine manipulating stretchable zone plate for adjusting the focus of ultrasonic beam.

**Figure 2 sensors-19-03819-f002:**
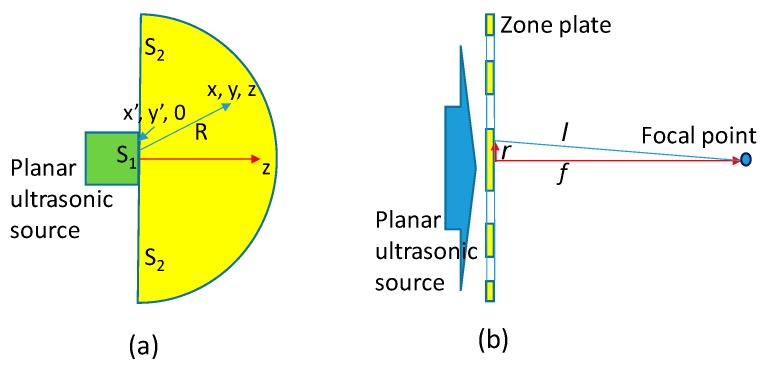
(**a**) Planar cylinder transducer enclosed in a baffle. (**b**) Diagram for locating the zone boundaries.

**Figure 3 sensors-19-03819-f003:**
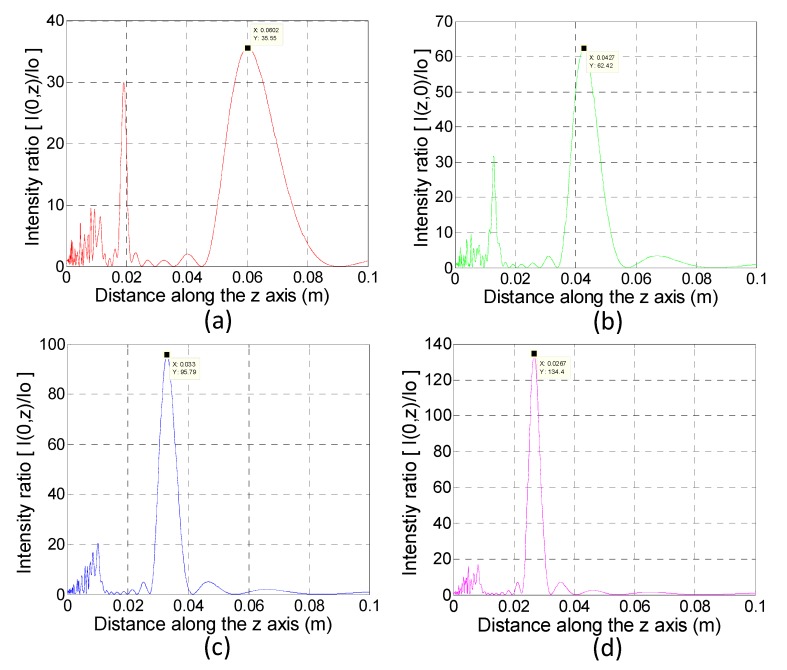
Calculated intensity ratios of four kinds of zone plate designs. (**a**) *n* = 5; (**b**) *n* = 7; (**c**) *n* = 9; (**d**) *n* = 11.

**Figure 4 sensors-19-03819-f004:**
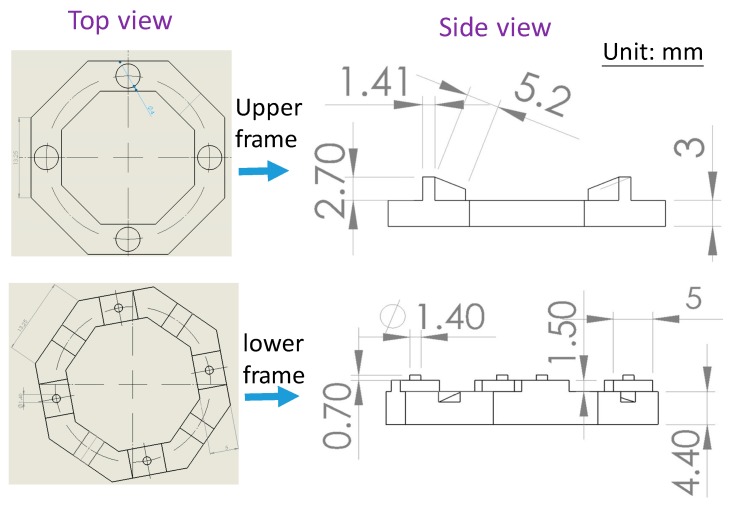
Design of the upper and lower frames by Solidworks.

**Figure 5 sensors-19-03819-f005:**
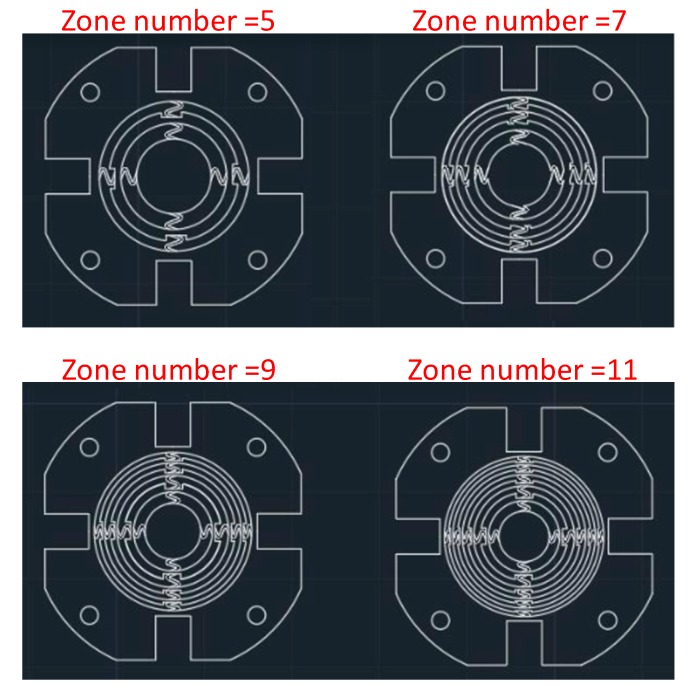
Photomask designs of the stretchable zone plate with zone number 5, 7, 9 and 11.

**Figure 6 sensors-19-03819-f006:**
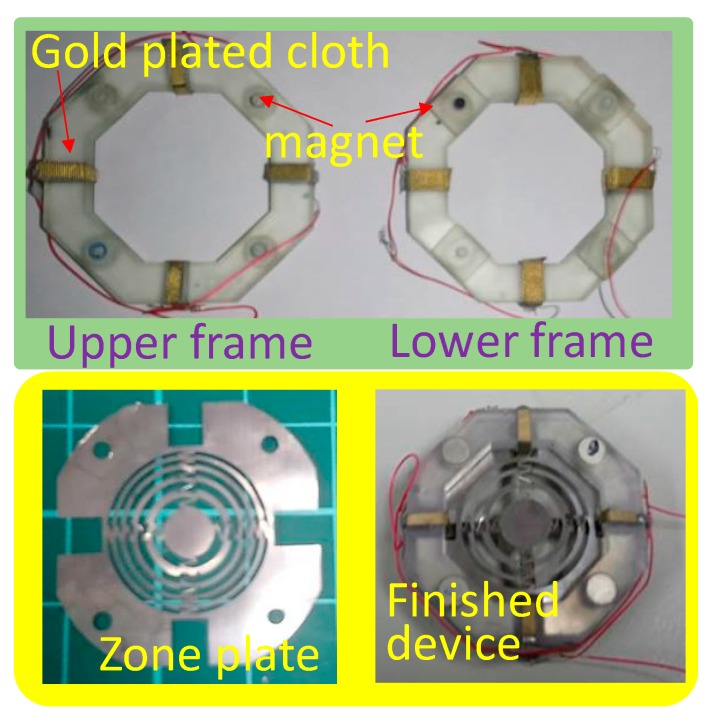
Fabrication results of the proposed device.

**Figure 7 sensors-19-03819-f007:**
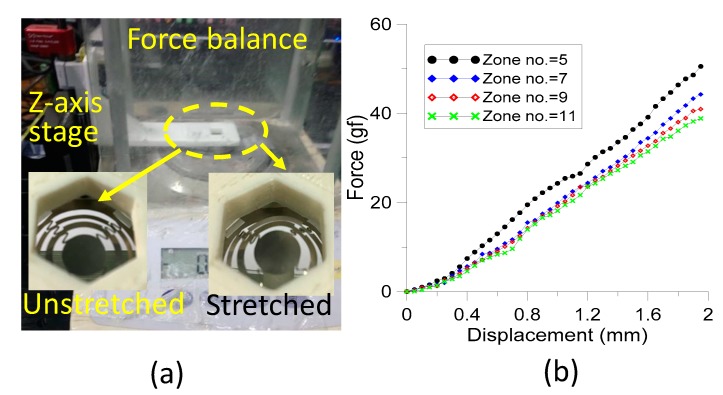
(**a**) Experimental setup for characterizing the stretchable zone plate. (**b**) The stretched displacement in the out-of-plane direction of the zone plate as a function of applied forces.

**Figure 8 sensors-19-03819-f008:**
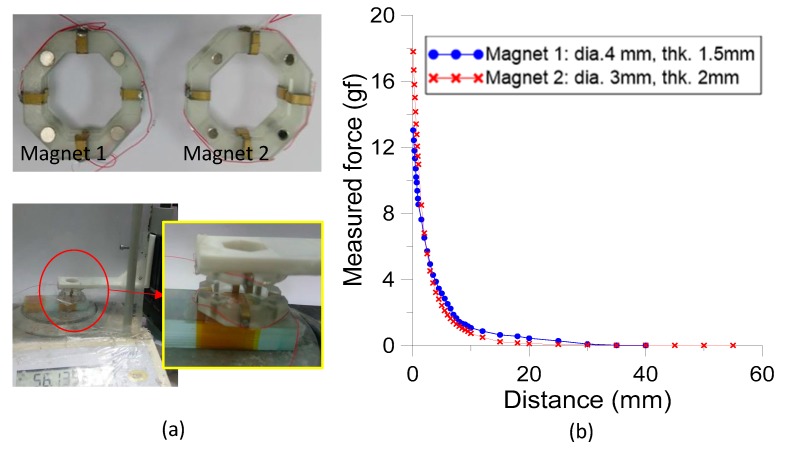
(**a**) Experimental setup for characterizing magnetic force between the frame pair. (**b**) Measured results of using different magnet pairs.

**Figure 9 sensors-19-03819-f009:**
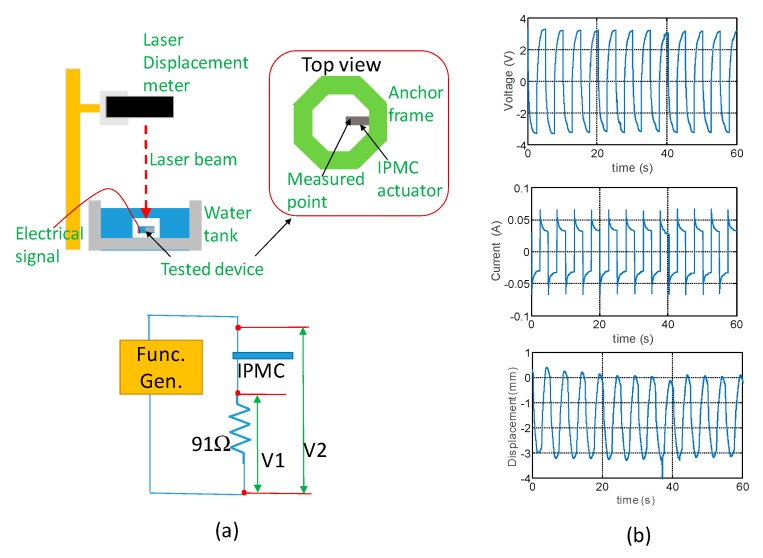
(**a**) Experimental setup for testing a single IPMC actuator performance in water. (**b**) Measured results of voltage drop across, current flowing through and displacement of the IPMC actuator.

**Figure 10 sensors-19-03819-f010:**
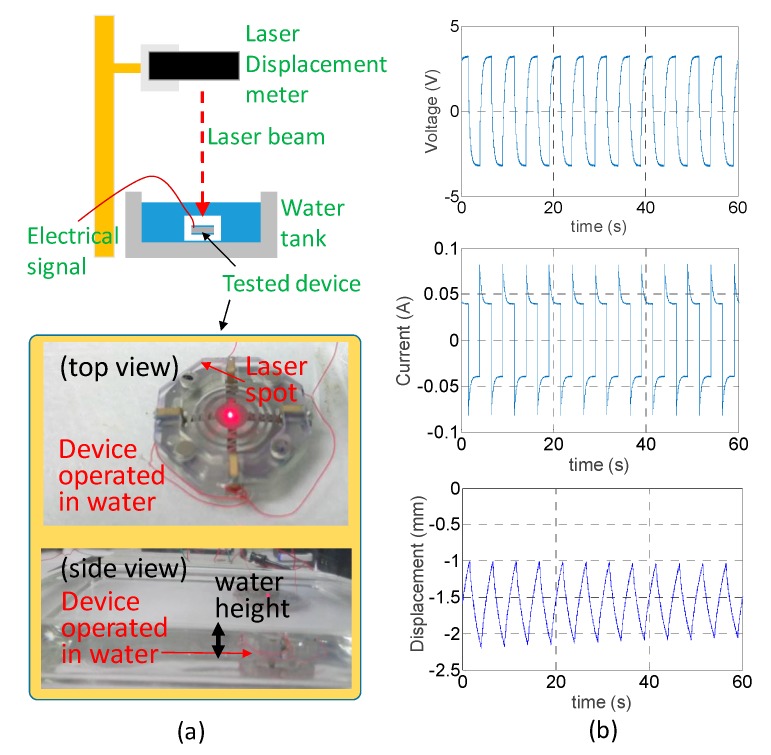
(**a**) Experimental setup for four IPMC actuator-constructed soft engine to stretch the zone plate in water. (**b**) Measured results of voltage drop across, current flowing through and displacement of the soft engine.

**Figure 11 sensors-19-03819-f011:**
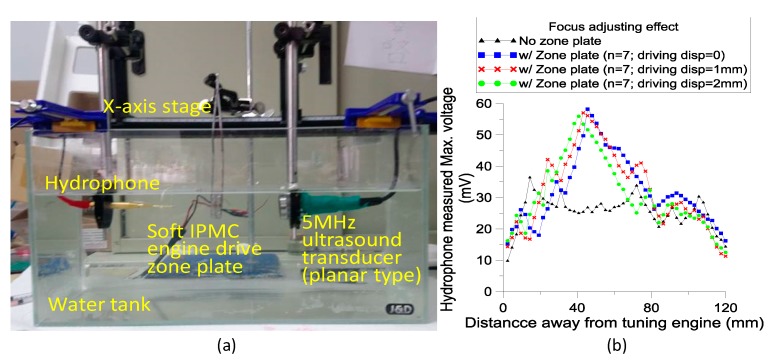
(**a**) Experimental setup for characterizing ultrasonic pressure while planar ultrasonic beam passing through the soft engine driven zone plate. (**b**) Measured maximum voltage output from hydrophone as a function of distance.

**Table 1 sensors-19-03819-t001:** Studied four designs of the zone plates with zone number 5, 7, 9 and 11.

Ring No.	Ring Boundary (mm)	Ring No.	Ring Boundary (mm)	Ring No.	Ring Boundary (mm)	Ring No.	Ring Boundary (mm)
1	4.26	1	3.601	1	3.177	1	2.874
2	6.024	2	5.092	2	4.493	2	4.064
3	7.378	3	6.237	3	5.503	3	4.977
4	8.519	4	7.201	4	6.354	4	5.747
5	9.525	5	8.051	5	7.104	5	6.426
		6	8.82	6	7.782	6	7.039
		7	9.526	7	8.406	7	7.603
				8	8.986	8	8.128
				9	9.531	9	8.621
						10	9.088
						11	9.531

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
