# Peer review of "An Out-of-Plane Operated Soft Engine Driving Stretchable Zone Plate for Adjusting Focal Point of an Ultrasonic Beam"

_sensors, 2019, doi:10.3390/s19183819_

Round 1
Reviewer 1 Report
The topic of focusing ultrasonic beams using soft actuators is surely an interesting one and the authors have communicated the background and the aspects involved in a thorough manner. Perhaps even too much so, as all the math provided is not really required to understand the principle of the device.
On the other hand, the aspects choosing the actuation parameters and some of the characteristics would benefit from additional details.
For instance:
The selection of the actuator material, the driving voltage range, etc should be explained, as these are not obvious. Especially considering that the Faradaic current observed may be related to water electrolysis, even if no gas evolution was mentioned in the manuscript. Soft ionic actuators, especially under load, tend to undergo creep. Moreover, IPMCs in particular have very poor position holding capabilities. These aspects should also be discussed, especially if the aim is to hold focus in a particular distance. A small soft actuator based engine should require some sort of feedback or control system to make it usable. The potential approaches should be discussed at least, if not tested out.The clarity of presentation can also be improved.
Namely the photos (alone) do a poor job explaining the experimental setup. For instance the ones in Fig 9. and 10. A scheme could be more informative, either added or even replacing the photos.
The manuscript also contains a relatively large number of terms, and sentences or parts of sentences that make no sense or are difficult to comprehend in their present form. For instance: “whose main chain is a linear hydrophobic structure and whose branch chains are sulfonic groups with adsorbed water molecules which are hydrophilic in nature“; ”stereographically fabricated“; “a prominent property is the stripe without appreciable fatigue“; “The actuation module of 4 IPMC beams bonded with the linkage structure was positioned to allow each IPMC beam arranged at slots of the lower farme.“; “The stiffness also exhibits a lower stiffness“; “IPMC in each cycle [36, 37]”. There are others, therefore, English should be improved.
Author Response
Thank you very much for your valuable time to process our paper (Manuscript: sensors-564526; Title: An Out-of-Plane Operated Soft Engine Driving Stretchable Zone Plate for Adjusting Focal Point of an Ultrasonic Beam). We have responded all the questions/comments that the reviewers asked and modified the manuscript (highlighted in yellow) as suggested in the revision.
Please review our revision and give us comments. We are looking forward to your positive response ASAP.

Reviewer 2 Report
An interesting article.
Some comments.
The abstract is fragmented and the What, Why, How and Accomplishment are not clear. Can the authors please lable Fig 5, which photo are zone number 5, 7, 9 and 11? Figure 5 caption typo error -> Photomask designs of the stretchable zone <platse> with zone number 5, 7, 9 and 11. IPMC is known for its time variable performance. How does the authurs ensure the repeatablility? What is its hysteresis? What are the details/specifications of the IPMC used?Author Response
Thank you very much for your valuable time to process our paper (Manuscript: sensors-564526; Title: An Out-of-Plane Operated Soft Engine Driving Stretchable Zone Plate for Adjusting Focal Point of an Ultrasonic Beam). We have responded all the questions/comments that the reviewers asked and modified the manuscript (highlighted in yellow) as suggested in the revision.
Please review our revision and give us comments. We are looking forward to your positive response ASAP.

Round 2
Reviewer 1 Report
The manuscript has been reasonably improved during the revision but there is still room for improvement. The authors have added discussion of soft actuator selection, but not quite the choice of IPMC over, say conducting polymer or CNT or some other nanostructured carbon-based ionic actuator.
Also, the choice of too driving voltage (causing electrolysis of water - potentially leading to explosion of H2/O2 mixture) was not justified.
Stability has been discussed slightly, but apparently without any experimental evidence.
